# The Role of Long Noncoding RNAs in Diabetic Alzheimer’s Disease

**DOI:** 10.3390/jcm7110461

**Published:** 2018-11-21

**Authors:** Young-Kook Kim, Juhyun Song

**Affiliations:** 1Department of Biochemistry, Chonnam National University Medical School, Hwasun 58128, Jeollanam-do, Korea; ykk@chonnam.ac.kr; 2Department of Anatomy, Chonnam National University Medical School, Hwasun 58128, Jeollanam-do, Korea

**Keywords:** LncRNAs, long noncoding RNAs, diabetic Alzheimer’s disease, synaptic dysfunction, inflammation, insulin resistance

## Abstract

Long noncoding RNAs (lncRNAs) are involved in diverse physiological and pathological processes by modulating gene expression. They have been found to be dysregulated in the brain and cerebrospinal fluid of patients with neurodegenerative diseases, and are considered promising therapeutic targets for treatment. Among the various neurodegenerative diseases, diabetic Alzheimer’s disease (AD) has been recently emerging as an important issue due to several unexpected reports suggesting that metabolic issues in the brain, such as insulin resistance and glucose dysregulation, could be important risk factors for AD. To facilitate understanding of the role of lncRNAs in this field, here we review recent studies on lncRNAs in AD and diabetes, and summarize them with different categories associated with the pathogenesis of the diseases including neurogenesis, synaptic dysfunction, amyloid beta accumulation, neuroinflammation, insulin resistance, and glucose dysregulation. It is essential to understand the role of lncRNAs in the pathogenesis of diabetic AD from various perspectives for therapeutic utilization of lncRNAs in the near future.

## 1. Introduction

Long noncoding RNAs (lncRNAs) are a class of noncoding RNAs [1] that regulate diverse physiological processes by controlling gene expression [2]. The sequences of most lncRNAs are longer than 200 nucleotides (nt) and do not have protein coding potential. They have been reported to be involved in various diseases, including cancer and cardiovascular and neurodegenerative diseases [3,4,5,6]. Among these, the roles of various lncRNAs have been recently reported in Alzheimer’s disease (AD) [7,8,9,10]. Considering the diverse roles and mechanism of action of lncRNAs, their utilization for the treatment of AD may be an appropriate approach for achieving high therapeutic effectiveness.

Diabetic AD is a newly emerging concept in the field of neurodegenerative diseases [11]. Risk factors for type 2 diabetes mellitus (T2DM), including impaired glucose homeostasis and insulin resistance, are believed to contribute to the onset and progression of AD [12,13,14]. With increasing age, metabolic failures trigger hyperglycemia and insulin resistance, leading to T2DM, and subsequently increase the risk of diabetic AD [15,16,17]. As previously mentioned, lncRNAs are broadly involved in AD pathogenesis through diverse mechanisms. Presently, the manipulation of diabetic AD-related lncRNAs in the brains of patients with AD is considered a novel therapeutic approach. Here, we broadly review the roles of lncRNAs in diabetes and AD, and suggest the therapeutic feasibility of lncRNAs against diabetic AD pathogenesis.

## 2. Diabetic Alzheimer’s Disease

The characteristic features of AD are memory dysfunction and cognitive decline, which are triggered by neurodegeneration resulting from amyloid beta (Aβ) accumulation and tau protein hyperphosphorylation in the brain [18]. Also, decades of research demonstrated that mutations in APP and the two presenilin genes (PS1 and PS2) can trigger the development of AD though misprocessing of APP [19,20,21]. Several studies have shown that diabetes could increase the risk of AD by influencing cognitive defects and insulin resistance in the brain [13,14]. One study revealed that T2DM enhances Aβ aggregation, which leads to severe Aβ pathology [22]. Another study reported that diabetes worsened amyloid plaque and neurofibrillary tangle formation in the brain of a triple transgenic AD mouse model [12].

Hyperglycemia is known to trigger neuronal death by boosting the production of advanced glycation products and by promoting glucose shunting [23]. Additionally, it was reported that T2DM caused by a high-fat diet is an important risk factor for cognitive dysfunction [24,25]. A study using insulin-resistant mice revealed memory deficits and impaired hippocampal insulin signaling in this animal model [26]. Further, AD transgenic mice fed a high-fat diet showed higher accumulation of higher accumulation of Aβ and tau protein in the cerebral cortex [27]. In a clinical study, patients with T2DM showed reduced hippocampal volume and impaired cognitive function compared to healthy subjects [28]. Previous studies suggested that diabetes is a strong risk factor for AD [11] and diabetes-induced AD has been emerging as a major pathogenic mechanism of AD [29,30]. Several studies demonstrated that hyperinsulinemia and insulin resistance triggers cognitive impairment and Aβ and amyloid precursor protein (APP) deposition [31,32,33,34]. Additionally, AD can be influenced by impaired insulin signaling [35], suggesting that tau hyperphosphorylation is regulated through insulin and the insulin-like growth factor signaling pathway [36]. The potential influence of diabetes on the onset and development of AD should be highlighted to prevent the global increase of AD. Further, considering previous broad findings, diabetic AD should be investigated in greater detail to find methods to attenuate the risk of AD onset and to understand the detailed mechanism of AD pathogenesis.

## 3. What is LncRNAs?

Noncoding RNAs (ncRNAs) are generally divided into two groups based on their sizes: small (< 200 nt) and long (> 200 nt) (Figure 1A). The small ncRNAs include transfer RNAs, small nucleolar RNAs, small nuclear RNAs, and microRNAs (miRNAs), whereas lncRNAs include heterogeneous classes of regulatory RNAs, such as antisense RNAs, enhancer RNAs, and long intergenic noncoding RNAs (lincRNAs) [37,38,39]. As regulatory ncRNAs, more lncRNAs exist in the human genome than miRNAs, but the function of lncRNAs is understood less well (Figure 1). Various loci in the genome encode lncRNAs: enhancers, promoters, introns of protein-coding genes, and intergenic regions [40]. Most lncRNAs are transcribed by RNA polymerase II and are 5′-capped but only a subset of lncRNAs are polyadenylated. They are spliced using the consensus splicing signals that are used for mRNAs [39,41]. The sequences of lncRNAs are characterized by their attenuated conservation across species, and fewer exons are observed than protein-coding transcripts [42,43]. Numerous studies have found that lncRNAs influence physiological processes by controlling gene expression at the transcriptional and posttranscriptional levels [44,45]. In cells, lncRNAs are localized in both the nucleus and cytoplasm [46], and they function through distinct mechanisms depending on their cellular localization [47]. In the nucleus, lncRNAs act as transcriptional activators or inhibitors in *cis* (to control neighboring genes on the same chromosome) or in *trans* (to control genes at distant regions of the same chromosome or on other chromosomes). These nuclear lncRNAs exert their functions via various mechanisms, including recruitment of chromatin modification complexes into essential genomic loci to influence DNA methylation [46,48], and act as transcriptional coactivators [47,49,50]. In contrast, lncRNAs in the cytoplasm mostly act as molecular decoys for proteins or miRNAs. In addition to their roles as transcriptional or post-transcriptional regulators, some lncRNAs have functions that are entirely independent of gene regulation, such as chromosome segregation, DNA damage repair, and genome stability [51,52].

LncRNAs have been implicated as modulators of various cellular mechanisms including regulation of RNA stability, cell cycle, and chromatin structure [53,54,55]. Dysregulation of lncRNAs has been implicated in the pathogenesis of many diseases, such as cancer [56,57] and cardiovascular diseases [58,59]. Moreover, recent studies have reported the importance of lncRNAs in the pathogenesis of diabetes, obesity, and neuronal diseases [60,61,62,63] (Table 1). Some lncRNAs, including myocardial infarction associated transcript (*MIAT*), metastasis-associated lung adenocarcinoma transcript 1 (*MALAT1*), and nuclear enriched abundant transcript 1 (*NEAT1*), influence neurodegenerative diseases [60] such as Huntington’s disease [61,62] and AD [64,65]. Importantly, *MALAT1* was shown to be involved in the regulation of synaptic density, hepatic steatosis, and insulin resistance [66,67]. Another lncRNA *H19* promotes neuroinflammation, and is also involved in insulin signaling and glucose uptake [68,69]. Thus, these lncRNAs are good candidate links between diabetes and AD that can be explored in future studies. Although there are considerably more lncRNAs than miRNAs in cells, fewer studies have been conducted on the functions of lncRNAs than on the functions of miRNAs, highlighting the importance of future studies on lncRNAs (Figure 1). In this review, we tried to identify important lncRNAs in the diabetic AD brain and the role of these lncRNAs focused on neurogenesis, neuroinflammation, synaptic dysfunction, Aβ accumulation, insulin resistance, and glucose homeostasis.

## 4. LncRNAs in AD

Current studies have reported that the expression patterns of lncRNAs appear disrupted in the brain, blood, and cerebrospinal fluid (CSF) of patients with AD compared to those in normal subjects. There have been reports of lncRNAs that are dysregulated in the brains of patients with AD [40]. Moreover, lncRNAs, including *RP11-462G22.1* and *PCA3*, were upregulated in exosomes derived from the CSF of patients with AD compared to normal subjects [93]. For example, in an in vitro AD model, the expression of antisense ubiquitin carboxyl-terminal hydrolase L1 (antisense *Uchl1*) increased, and this in turn led to an increase in the synthesis of the UCHL1 protein, a deubiquitinating enzyme involved in AD pathogenesis, at the posttranscriptional level [70]. Although several studies have reported the relationship between the altered expression of lncRNAs and AD progression, more efforts are necessary to understand the regulatory mechanisms of specific lncRNAs in the brain, blood, and CSF of patients with AD. Moreover, the study of lncRNAs involved in the link between the onset of AD and T2DM and involving Aβ accumulation or neuroinflammation is required to elucidate the related mechanisms and to find the solution to inhibit these pathogeneses. Hence, we summarize the known lncRNAs involved in these processes below.

## 5. LncRNAs Control Neurogenesis

The regulation of neurogenesis is an important issue for AD as impaired neurogenesis leads to cognitive impairments through neuronal loss and synaptic dysfunction [94,95]. Neural stem cells (NSCs), which have the ability of self-renewal, ultimately differentiate into neurons or neuroglia depending on the action of numerous trophic factors [96,97]. Neurogenesis occurs mainly in the anterior part of the subventricular zone and in the subgranular zone of the hippocampus [96,98]. In the brain, hippocampal neurogenesis has a critical role in memory function and is influenced by growth factors including brain-derived neurotrophic factor, insulin growth factor-1, and vascular endothelial growth factor [99]. Several lncRNAs play roles in the glial and neuronal differentiation of NSCs [100,101]. One study listed the lncRNAs that are associated with neuronal differentiation [60], and others have identified several lncRNAs involved in the regulation of neural lineage specification and neuron-glia fate switching [102,103,104]. Further, several lncRNAs are involved in hippocampal neuronal maturation and oligodendrocyte differentiation and maturation by modulating the epigenetic status of protein-coding genes [41,103]. Additionally, two lncRNAs, *NEAT1* and *MALAT1*, can upregulate neurogenesis by controlling neurite outgrowth and neuronal fate [102,104,105]. Neurite outgrowth is an essential process of neuronal differentiation and also plays an important role in neuronal regeneration and injury response [106]. It is a biological mechanism involving complicated regulation of gene expression and signal transduction [106]. Moreover, it was reported that lncRNAs, including *RMST*, are involved in neuronal differentiation [75]; *RMST* could increase neuronal differentiation by physically interacting with SOX2 (a transcription factor known to regulate neural fate) and by binding to promoter regions of genes encoding neurogenic transcription factors. Considering this evidence for the role of lncRNAs in neural differentiation and maturation and glial differentiation, further studies on lncRNAs are required to investigate the regulation of neurogenesis in diabetic AD, given that diabetes leads to impaired neurogenesis and aggravates the progression of AD [107]. The modulation of neurogenesis-associated lncRNAs in diabetic AD may be an important key to improve neurogenesis and ultimately to inhibit memory dysfunction.

## 6. LncRNAs Regulate Synaptic Dysfunction in AD

Synaptic dysfunction is a critical issue in the pathogenesis of AD since it drives cognitive decline [108,109]. Synaptic dysfunction occurs in various cortical circuits of the brain, from the entorhinal cortex to the hippocampus [110,111]. Impaired synaptic plasticity results in a decreased number of synapses between neurons, and neuronal loss, which leads to cognitive dysfunction in AD [112]. Previous studies have shown that the lncRNA *BC1*/*BC200* influences synaptic plasticity [113,114]. It was reported that the expression of *BC200* was lower in the brains of patients with AD than that in the brains of normal subjects, and that structural disturbance in the *BC200* RNA resulted in the inhibition of dendritic delivery [65]. Inadequate translational modulation was observed in *Bc1* knockout mice, and this triggered neuronal hyper-excitability and the blockade of extracellular signal-regulated kinase-mitogen-activated protein kinase kinase signaling [115]. Another study found that *BC1* RNA could contribute to the regulation of striatal γ-aminobutyric acid (GABA) synapses through dopamine receptor D2 in AD [116]. Furthermore, the lncRNA *Gomafu* is expressed throughout the brain [117], and its dysregulation leads to synapse dysfunction by triggering aberrant splicing of ERRB4 [118]. Thus, several lncRNAs could regulate synaptic plasticity in AD, suggesting that it would be plausible to reverse impaired synaptic plasticity by modulating these lncRNAs. Given that the improvement of impaired synaptic plasticity could resolve memory dysfunction in diabetic AD, the identification of lncRNAs related to synaptic plasticity is essential to find a therapeutic solution.

## 7. LncRNAs Modulate Aβ Accumulation in AD

APP is cleaved by the β-site amyloid precursor protein cleaving enzyme 1 (BACE1 or β-secretase 1) and γ-secretase, leading to the production of Aβ peptides [119]. Abnormal Aβ clearance and increase in BACE1 activity promoted Aβ accumulation and aggravated AD progression [119]. Several lncRNAs were shown to regulate the expression of APP and BACE1 in the brains of patients with AD [77,120]. For example, neuronal sortilin-related receptor gene (*SORL1*) is known to influence the cleavage of APP and lead to the inhibition of Aβ formation in the brains of patients with AD [120]. The lncRNA *51A*, mapped at an antisense orientation to intron 1 of *SORL1* [121], was overexpressed in the brain of patients with AD and was found to enhance Aβ formation [77]. The increase in 51A expression markedly decreased SORL1 levels, possibly by controlling the splicing pattern of the *SORL1* transcript [77]. In the brains of patients with AD, the mRNA levels of BACE1, which contributes to the development of AD, are reduced in the cortex, dorsal hippocampus, and ventral hippocampus, but not in the cerebellum [26,122]. Loss of BACE1 triggers behavioral deficits, such as memory loss [123] and loss of synaptic plasticity [124]. The levels of the lncRNA *BACE1-AS*, which is transcribed in an antisense orientation to *BACE1*, is considerably increased in AD, and its overexpression could elevate Aβ_1-42_ levels [7,125]. Further, *BACE1-AS* could regulate the mRNA levels of BACE1, which increases Aβ production [26]. Moreover, this lncRNA binds to HuD, an RNA-binding protein associated with learning and memory function, and influences APP cleavage [79]. In the neuroblastoma cell line SH-SY5Y, an in vitro model of AD, the lncRNAs *17A* and neuroblastoma differentiation marker 29 (*NDM29*) were significantly upregulated; this in turn promoted the accumulation of Aβ [76,81]. An RNA polymerase III-transcribed ncRNA regulated by an extragenic type-3 promoter [126], *NDM29*, is also involved in the synthesis of APP, leading to increased Aβ secretion [127,128]. Other studies have also shown that *NDM29* contributes to Aβ formation by regulating its cleavage [129,130]. This evidence indicates that lncRNAs contribute to the accumulation of Aβ through diverse mechanisms and ultimately affect the pathogenesis of AD. It is necessary to elucidate the mechanisms between candidate lncRNAs and Aβ accumulation to attenuate Aβ accumulation in the brain which may lead to a therapeutic approach against AD.

## 8. LncRNAs Contribute to the Regulation of Neuroinflammation in AD

Inflammatory signaling is an important factor for brain homeostasis, repair, and neuroprotection [105]. In AD, the inflammatory response is a key hallmark because it aggravates the pathogenesis of AD [131]. The accumulation of Aβ leads to excessive oxidative stress and triggers severe neuroinflammation [128,132]. Several studies have reported that lncRNAs influence immune responses [133,134]. Interestingly, many genes encoding immune-related lncRNAs including IL1β-RBT46 [135], LincR-Ccr2-5’AS [136], and lnc-IL7R [137], are located close to immune-responsible protein coding gene clusters, suggesting that these lncRNAs and proteins are under a common signaling pathway. It was demonstrated that the lncRNA *LincRNA-Cox2* activates immune genes in macrophages and mediates the activation of NF-κB [83]. This lncRNA works by binding with the RNA-binding proteins HNRNPA/B and HNRNPA2/B1 and by enhancing the expression of interleukin (IL)-6 through toll-like receptors. Further, defects in the lncRNA *Lethe* trigger the upregulation of *NFKBIA* and *NFKB2*, and lead to the blockage of NF-κB activation [82]. Additionally, lncRNA H19 promotes neuroinflammation by regulating M1 microglial polarization through histone deacetylases [68]. In the neuroinflammatory response, *NEAT1*, the lncRNA known to regulate the formation of the nuclear paraspeckle body, also stimulated the expression of IL-8, which is associated with cognitive dysfunction [85,138,139]. Moreover, *NDM29* was shown to be involved in the proinflammatory response in patients with AD [81]. Furthermore, *17A* decreased the transcription of the GABA B2 receptor, which remarkably impaired the GABAB signaling pathway and enhanced neuroinflammation in AD brains [76]. As mentioned above, neuroinflammation is an important cause in neuronal damage [140,141,142] and simultaneously, the identification and modulation of neuroinflammation-related lncRNAs may be a good approach to treat neuropathies caused by diabetes-induced AD.

## 9. LncRNAs Control Insulin Resistance and Glucose Homeostasis

Insulin resistance is characterized by impaired insulin action in maintaining glucose homeostasis [143]. Given that insulin resistance in the brain is a crucial risk factor for AD and aggravates the pathogenesis of AD [144,145], further studies on the relationship between AD and insulin resistance in the brain are warranted to investigate possible therapeutic approaches against AD. Since glucose dysregulation in the brain is expected to be one of the causes of AD [146], it is interesting that several studies have revealed the regulation of glucose homeostasis by lncRNAs [147,148]. Further, numerous other studies have found that lncRNAs play crucial roles in the progression of T2DM through processes including insulin resistance and impaired glucose homeostasis [149,150,151]. One study reported that interference of the lncRNA maternally expressed gene 3 (*Meg3*) reduced the levels of triglycerides and attenuated impaired glucose tolerance in a high-fat diet fed obesity mouse model [88]. Additionally, *Meg3* was shown to promote insulin resistance in a mouse model of type 1 diabetes mellitus (T1DM) and T2DM [152]. Knockdown of *Meg3* reduced insulin synthesis and its inactivation decreased the synthesis and secretion of insulin, leading to glucose tolerance. Furthermore, a study of patients with diabetes revealed an association between a SNP (rs941576), located on an intron of *MEG3*, with T1DM [153]. In addition, H19 lncRNA, which was decreased in human diabetic subjects and insulin resistant mice, could regulate glucose homeostasis by acting as a molecular sponge for let-7 miRNAs [89]. Plasmacytoma variant translocation 1 (*PVT1*), a lncRNA stimulated by glucose, was shown to be associated with T1DM and T2DM [147]. Further, the expression of *MIAT* was upregulated in the vascular endothelial cells of diabetic rats, as well as those of patients with diabetes, under high-glucose conditions [154]. Another study reported that the lncRNA steroid receptor RNA activator (*SRA*) was shown to exist in a ribonucleoprotein complex bound with the trithorax group or polycomb repressive complex 2 and modulate gene expression [155]; *SRA* could increase the expression of the insulin receptor and enhance insulin signaling by blocking the phosphorylation of c-Jun, N-terminal kinase, and p38 mitogen-activated protein kinase [8]. The mature adipocytes of *SRA* knockout mice exhibited decreased insulin sensitivity resulting from reduced phosphorylation of insulin receptor substrate-1 and decreased glucose levels in the brains of high-fat diet fed mice [8]. Furthermore, it was shown that *MALAT1* expression was increased in the liver of obese mice and promoted hepatic insulin resistance by activating the stability nuclear sterol regulatory element binding transcription factor 1c (SREBP-1C) [67]. This lncRNA was also implicated in its association with the pathogenesis of diabetes-related microvascular diseases [156], and its knockdown suppressed the viability of endothelial cells [157] and inhibited the expression of inflammatory marker genes [84]. Interestingly, in addition to its roles in insulin resistance, the function and working mechanism of *MALAT1* were also studied in neuronal process as described above [66,105], implying that this lncRNA might be an example of lncRNA linking both processes. Thus, an important study in the future will be the identification of the role of *MALAT1* in diabetic AD patients or related experimental models. Since insulin resistance and impaired glucose homeostasis in the brain lead to aggravation of AD progression [158], the lncRNAs associated with these processes merit further investigation to identify therapeutic approaches against AD.

## 10. Conclusions

As the research field of lncRNAs continues to expand worldwide, the knowledge of how lncRNAs act at a cellular level in diabetic AD is also gradually increasing. In this review, we summarized the relationship between lncRNAs and diabetic AD pathogenesis based on recent evidence. Several lncRNAs were associated with diverse processes in the brain underlying the pathogenesis of diabetic AD, including impaired neurogenesis, synaptic dysfunction, Aβ accumulation, neuroinflammation, insulin resistance, and impaired glucose homeostasis. To our knowledge, most lncRNAs related to diabetic AD have been discussed in this review, but investigation of lncRNAs on diabetic AD is still in the early stage. We highlight that lncRNAs are desirable candidates for future studies on diabetic AD biomarkers and the information about lncRNAs in this review may help suggest potential diabetic AD treatments. The comprehensive understanding of lncRNA in AD and diabetes may help us attenuate AD pathology. As more discoveries on the changes in lncRNA expression associated with diabetic AD will be made, we can utilize those lncRNAs as the biomarkers for the disease status of diabetic AD. Furthermore, the modulation of lncRNA expression using genome engineering technologies will enable us to apply clinically relevant lncRNAs into the therapy of diabetic AD. Hence, we suggest that further studies regarding the roles of lncRNAs in diabetic AD are essential to identify advanced therapeutic approaches against the disease.

## Figures and Tables

**Figure 1 jcm-07-00461-f001:**
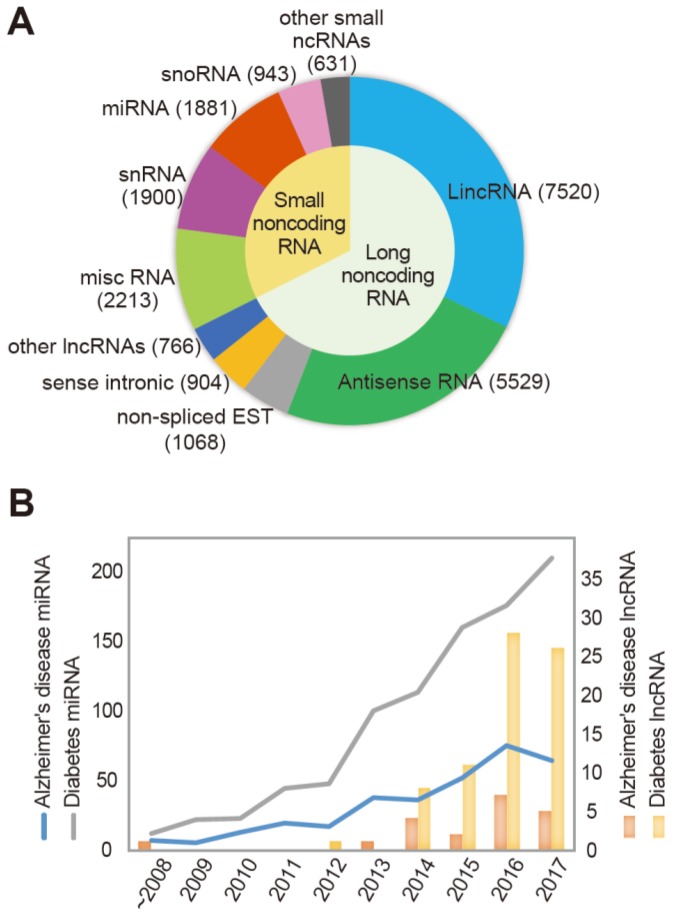
The classification of noncoding RNAs (ncRNAs) and the number of papers published on the study of ncRNAs in Alzheimer’s disease (AD) and diabetes. (**A**) The number of ncRNAs identified in humans. The classification of ncRNAs is based on GENCODE (Version 26, October 2016) [92]. The number of ncRNAs in each group is indicated in parentheses. (**B**) The number of research papers on long ncRNAs (lncRNAs) and microRNAs (miRNAs) in AD and diabetes published in the past 10 years is shown. The papers were located in PubMed and review papers were excluded. Note that although the number of lncRNAs, including long intergenic ncRNAs (lincRNAs), antisense RNAs, and sense intronic RNAs, is larger than that of regulatory small RNAs, only a few papers have been published on lncRNAs in AD and diabetes compared to those published on miRNAs. EST: expressed sequence tag, snRNA: small nuclear RNA, snoRNA: small nucleolar RNA.

**Table 1 jcm-07-00461-t001:** List of lncRNAs related to neurogenesis and synaptic function, Aβ accumulation, immune response, insulin signaling and glucose homeostasis. Only those with known mechanisms were selected. The genomic loci of lncRNAs are shown based on the genomic coordinates at hg19 genome assembly for humans or at mm10 assembly for mice, respectively.

LncRNA	Genomic Locus (hg19 or mm10)	Function	Mechanism	Reference
**Neurogenesis and synaptic function**				
antisense Uchl1	chr5:66626495–66676497 (mouse)	Possibly involved in brain function and neurodegenerative diseases through the regulation of Uchl1	Antisense Uchl1 RNA is required for the association of the overlapping Uchl1 mRNA to activate polysomes for translation	[70]
Dali	chr1:42750712–42752886 (mouse)	Regulates neural differentiation	Interacts with the DNMT1 DNA methyltransferase in mouse and human and regulates DNA methylation status of CpG island-associated promoters in *trans*	[71]
Evf2	chr6:6820543–6871592 (mouse)	Evf2 mouse mutants had fewer GABAergic interneurons in the early postnatal hippocampus and dentate gyrus	Recruited DLX and MECP2 transcription factors to important DNA regulatory elements in the Dlx5/6 intergenic region and controlled Dlx5, Dlx6 and Gad1 expression	[72]
MALAT1	chr19:5795690–5802671 (mouse)	Regulates synaptic density	Modulates the recruitment of serine/arginine-rich (SR) family pre-mRNA-splicing factors to the transcription site	[66]
Miat	chr5:112213228–112228948 (mouse)	Involved in neurogenic commitment	Manipulation of Miat triggers pleiotropic effects on brain development and aberrant splicing of Wnt7b	[60]
Pnky	chr4:22490548–22493126 (mouse)	Regulates neurogenesis from embryonic and postnatal neural stem cell populations	Pnky interacts with the splicing regulator PTBP1	[73]
PVT1	chr15:62037986–62250976 (mouse)	Decreased by autophagic inhibition in diabetes	PVT1-mediated autophagy may protect hippocampal neurons from impairment of synaptic plasticity and apoptosis, and ameliorate cognitive impairment in diabetes	[74]
RMST	chr12:97858799–97927544 (human)	Regulation of neural stem cell fate	RMST is required for the binding of SOX2 to promoter regions of neurogenic transcription factors	[75]
**Aβ accumulation**				
17A	chr9:101258962–101259132 (human)	Enhances the secretion of Aβ and the Aβ_x-42_/Aβ_x-40_ peptide ratio	Induces the synthesis of an alternative splicing isoform that abolishes GABA B2 intracellular signaling	[76]
51A	chr11:121323765–121324036 (human)	Associated with impaired processing of amyloid precursor protein leading to increased Aβ formation	Drives a splicing shift of SORL1 and decreased the synthesis of SORL1 variant	[77]
BACE1-AS	chr11:117162062–117162886 (human)	Downregulation attenuates the ability of BACE1 to cleave APP and delays the induction of senile plaque formation	BACE1-AS forms RNA duplexes and increases the stability of BACE1 mRNA.	[78]
BACE1-AS	chr11:117162062–117162886 (human)	Partly complements BACE1 mRNA and enhances BACE1 expression	Associated with HuD	[79]
BACE1-AS	chr11:117162062–117162886 (human)	Directly implicated in the increased abundance of Aβ_1-42_ in Alzheimer’s disease	Elevated BACE1-AS increases BACE1 mRNA stability and generates additional Aβ_1-42_	[26]
BC1	chr7:144,914,470–144,914,623 (mouse)	Involved in Aβ aggregation and protection against spatial learning and memory deficits	Induces APP mRNA translation via association with a fragile X syndrome protein (FMRP)	[80]
NDM29	chr11:8,960,365–8,960,710 (human)	In patients affected by neurodegenerative diseases, synthesis of NDM29 is increased	NDM29-dependent cell maturation induces APP synthesis, leading to the increase of Aβ secretion and the concomitant increment of Aβ_x-42_/Aβ_x-40_ ratio	[81]
**Immune response**				
H19	chr11:2016406–2019105 (human)	Promotes neuroinflammation	Drives HDAC1-dependent M1 microglial polarization	[68]
Lethe	chr4:132219893–132220589 (mouse)	Regulates inflammatory signaling	Interacts with NF-κB subunit RelA to inhibit RelA DNA binding and target gene activation	[82]
lincRNA-Cox2	chr1:150159043–150164948 (mouse)	Mediates both the activation and repression of distinct classes of immune genes	Transcriptional repression of target genes is dependent on interactions of lincRNA-Cox2 with heterogeneous nuclear ribonucleoprotein A/B and A2/B1	[83]
MALAT1	chr11:65265209–65273987 (human)	Regulates glucose-induced up-regulation of inflammatory mediators IL-6 and TNF-α	Through activation of SAA3 expression	[84]
NEAT1	chr11:65190269–65213011 (human)	Facilitates the expression of antiviral genes including cytokines such as interleukin-8, and plays an important role in the innate immune response	NEAT1 induction relocates SFPQ from the IL8 promoter to the paraspeckles, leading to transcriptional activation of IL8	[85]
**Insulin signaling and glucose homeostasis**				
GM10768	chr19:43838803–43840845 (mouse)	Overexpression of Gm10768 activates hepatic gluconeogenesis while knockdown of Gm10768 improves glucose tolerance and hyperglycemia	Gm10768 sequestrates miR-214 to relieve its suppression on ATF4, a positive regulator of hepatic gluconeogenesis	[86]
H19	chr11:2016406–2019105 (human)	H19 depletion results in impaired insulin signaling and decreased glucose uptake	PI3K/AKT-dependent phosphorylation of KSRP promotes biogenesis of let-7 miRNA, and let-7 in turn, downregulates H19	[69]
H19	chr7:142575530–142578146 (mouse)	Regulates the expression of gluconeogenic genes and hepatic glucose output	H19 depletion impaired insulin signaling and increased nuclear localization of FoxO1	[87]
MALAT1	chr11:65265209–65273987 (human)	Promotes hepatic steatosis and insulin resistance	Interacted with SREBP-1c to stabilize nuclear SREBP-1c protein	[67]
MEG3	chr12:109540996–109571729 (mouse)	MEG3 suppresses insulin-stimulated glycogen synthesis in primary hepatocytes	MEG3 overexpression increases FoxO1, G6pc, Pepck mRNA expressions and hepatic gluconeogenesis	[88]
PCGEM1	chr2:193614571–193641625 (human)	Promotes glucose uptake for aerobic glycolysis, coupling with the pentose phosphate shunt to facilitate biosynthesis of nucleotide and lipid, and generates nicotinamide adenine dinucleotide phosphate (NADPH) for redox homeostasis	Binds directly to target promoters, physically interacts with c-Myc, promotes chromatin recruitment of c-Myc, and enhances its transactivation activity	[89]
Risa	chr10:63339203–63340913 (mouse)	Regulates insulin sensitivity	Overexpression of Risa decreases autophagy while knockdown of Risa up-regulates autophagy	[90]
SRA	chr18:36667187–36670311 (mouse)	SRA KO mice are resistant to high fat diet-induced obesity, with decreased fat mass and increased lean content, and more insulin sensitivity	Functions as a transcriptional coactivator of PPARγ and promotes adipocyte differentiation in vitro	[8]
βlinc1	chr2:147204578–147212616 (mouse)	Deletion of βlinc1 results in defective islet development and disruption of glucose homeostasis in adult mice	Regulates a number of islet-specific transcription factors located in the genomic vicinity of βlinc1	[91]

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
