# Peer review of "The Role of Long Noncoding RNAs in Diabetic Alzheimer’s Disease"

_jcm, 2018, doi:10.3390/jcm7110461_

Reviewer 1 Report

This paper on the role of long noncoding RNAs in diabetic Alzheimer’s disease is very interesting to the readers. However, authors are trying to connect 3 aspects, LncRNAs, diabetes, and Alzheimer’s disease, which were not clearly proven. Hence, this paper should be catagorized as prospective paper.

Yes, AD can be referred as Type III diabetes. And LncRNAs could be involved in AD. But there is no evidence on the LncRNA in diabetic AD, where inhalation of insulin only worked in few percentages of patients with AD, not all. Secondly, there was no separation of diabetic AD verse other AD.

If authors could find other papers on the segregation of AD patients with diabetic complication and others, and LncRNA were valid in those diabetic AD patients, I will highly recommend this paper as review paper. At current state of research, this paper could miss lead readers.

In conclusion, this paper should be catagorized as  prospective paper.

Author Response

Thanks for the suggestion and we fully agree with the opinion of the reviewer. As the reviewer said, there is still no direct and clear evidence which can link all those three aspects. However, we believe that discoveries will be made in this area through further researches, and we expect that the reviewer would agree with us. We discussed with the editor that our paper is more proper to be categorized as a prospective section as the reviewer said, and the editor helped us to change the paper type. Now the paper is categorized as a perspective paper.

Reviewer 2 Report

The review is well reading. Authors have covered Alzheimer’s disease, insulin resistance and lncRNAs well.

I have following points to make:

1.       Include the reference - Long noncoding RNAs and Alzheimer’s disease (Qiong Luo and Yinghui Chen; Clin Interv Aging. 2016; 11: 867–872).

2.       “Also, a recent study suggested that AD is triggered by impairment of the amyloid precursor protein (APP) metabolism, β- and γ-secretase mutation and presenilin (PS) mutations [18].”   this suggestion is not recent….so rephrase this.

3.       Include BC200 and Nat-Rad 18 to the table.

4.       Authors have discussed lncRNAs involved in neurodegeneration and insulin resistance separately- are there any overlap of lncRNAs involved in both the processes? Explain that.

5.       Consider making a section : how the lncRNAs can be therapeutic targets.

Author Response

Reviewer 2

The review is well reading. Authors have covered Alzheimer’s disease, insulin resistance and lncRNAs well.

I have following points to make:

1. Include the reference - Long noncoding RNAs and Alzheimer’s disease (Qiong Luo and Yinghui Chen; Clin Interv Aging. 2016; 11: 867–872).

à We added the reference in our revised manuscript as the reviewer suggested.

2. “Also, a recent study suggested that AD is triggered by impairment of the amyloid precursor protein (APP) metabolism, β- and γ-secretase mutation and presenilin (PS) mutations [18].” This suggestion is not recent….so rephrase this.

à We modified this sentence as shown below following the reviewer’s suggestion.

“Also, a previous study suggested that AD is triggered by impairment of the amyloid precursor protein (APP) metabolism, β- and γ-secretase mutation and presenilin (PS) mutations [19].”

3. Include BC200 and Nat-Rad 18 to the table.

à In the table, we only included those lncRNAs with their working mechanism has been identified. To our knowledge, there is only an experiment which measured the expression of Nat-Rad18 and its counterpart Rad18 (Parenti et al, Eur J Neurosci, 2007). Moreover, the studies for BC200 only identified their expression pattern in the AD or metabolic diseases. If the reviewer recommends any related paper to us, we will gladly include those lncRNAs to the table.

4. Authors have discussed lncRNAs involved in neurodegeneration and insulin resistance separately- are there any overlap of lncRNAs involved in both the processes? Explain that.

à MALAT1 is the only lncRNA with identified working mechanism which was studied both in neuronal process and insulin resistance to date. So we added the following sentences into the section 9.

“Interestingly, in addition to its roles in insulin resistance, the function and working mechanism of MALAT1 were also studied in neuronal process as described above [64,81], implying that this lncRNA might be an example of lncRNA linking both processes. Thus, an important study in the future will be the identification of the role of MALAT1 in diabetic AD patients or related experimental models.”

5. Consider making a section: how the lncRNAs can be therapeutic targets.

à Because there is still not so many studies which used lncRNAs as therapeutic targets, we think that it is not proper to describe a full section for the subject. Therefore, we added several sentences as shown below in the conclusion section to emphasize the subject following the reviewer’s suggestion.

“As more discoveries on the changes in lncRNA expression associated with diabetic AD will be made, we can utilize those lncRNAs as the biomarkers for the disease status of diabetic AD. Furthermore, the modulation of lncRNA expression using genome engineering technologies will enable us to apply clinically relevant lncRNAs into the therapy of diabetic AD.”

Round  2

Reviewer 2 Report

It can be accepted now.

Author Response

Dear reviewer

We tried to correct our manuscript following your comments.

We hope to be better than before.

Sincerely, thank you for your consideration

Best regards, Juhyun Song/ Young kook Kim